# Cell Autonomous Dysfunction and Insulin Resistance in Pancreatic α Cells

**DOI:** 10.3390/ijms20153699

**Published:** 2019-07-28

**Authors:** Norikiyo Honzawa, Kei Fujimoto, Tadahiro Kitamura

**Affiliations:** 1Division of Diabetes, Metabolism and Endocrinology, Department of Internal Medicine, Jikei University School of Medicine, 3-25-8 Nishishinbashi, Minato-ku, Tokyo 105-8461, Japan; 2Metabolic Signal Research Center, Institute for Molecular and Cellular Regulation, Gunma University, 3-39-15 Showa-machi, Maebashi, Gunma 371-8512, Japan

**Keywords:** pancreatic α cells, insulin resistance, glucagon

## Abstract

To date, type 2 diabetes is considered to be a “bi-hormonal disorder” rather than an “insulin-centric disorder,” suggesting that glucagon is as important as insulin. Although glucagon increases hepatic glucose production and blood glucose levels, paradoxical glucagon hypersecretion is observed in diabetes. Recently, insulin resistance in pancreatic α cells has been proposed to be associated with glucagon dysregulation. Moreover, cell autonomous dysfunction of α cells is involved in the etiology of diabetes. In this review, we summarize the current knowledge about the physiological and pathological roles of glucagon.

## 1. Physiological Roles of Pancreatic Islet Hormones on Glucose Metabolism

Neuronal and hormonal factors in the body strictly control blood glucose levels. The development of diabetes is attributed to the breakdown of the blood glucose control mechanism [1]. In 1921, Banting and Best discovered insulin, a hormone that lowers blood glucose levels in the body [2]. Insulin is secreted from pancreatic β cells and binds to insulin receptors (IRs) in classical insulin target organs, such as the liver, adipose tissue, and skeletal muscle [3]. In the adipose tissue and skeletal muscle, glucose transporter (GLUT)4 translocate from cytosol to the outer membrane via the phosphatidylinositol-3 kinase/protein kinase B (PI3K/Akt) pathway-dependent manner, which is a downstream signaling pathway of IR. Therefore, insulin promotes glucose uptake in these cells via GLUT4, thus resulting in a decrease in blood glucose levels [4]. Conversely, both GLUT2 and GLUT4 are involved in glucose uptake in the liver [5].

In 1923, Kimball and Murlin discovered glucagon from pancreas [6]. Glucagon is a polypeptide hormone comprising 29 amino acids, with a molecular weight of 3485 Da; it is secreted from α cells in the pancreatic islets of Langerhans but is also partially secreted from endocrine cells in the stomach and intestine. The primary physiological actions of glucagon are glycogenolysis, gluconeogenesis, and amino acid metabolism. After glucagon binds to glucagon receptors (GRs) in the liver, the acetylation of the cyclic adenosine monophosphate response element binding protein-regulated transcription coactivator (CRTC)2 and forkhead box protein O (FOXO)1 induce the expression of phosphoenolpyruvate carboxykinase (PEPCK) and glucose 6-phosphatase(G6-Pase) the rate-limiting enzymes of gluconeogenesis to promote gluconeogenesis while suppressing glycolysis [7,8,9]. Furthermore, glucagon promotes the degradation of glycogen to glucose by enhancing the phosphorylation of glycogen phosphorylase (GPase) via the cyclic adenosine monophosphate-protein kinase A (cAMP-PKA) pathway (glycogenolysis), thereby increasing the blood glucose levels [9]. Conversely, glucocorticoids increase blood glucose levels by promoting the glycation of proteins (deamination group) in the muscle cells and gluconeogenesis in the liver. Moreover, pancreatic islet hormones, such as somatostatin (secreted from pancreatic δ cells), pancreatic polypeptide (secreted from pancreatic polypeptide cells), and ghrelin (secreted from pancreatic ε cells), play a role in metabolic homeostasis, including blood glucose regulation. In healthy individuals, these hormones accurately regulate the blood glucose level at 90–120 mg/dL. The dysregulation of these hormones is involved in the onset and exacerbation of diabetes.

## 2. Cell Autonomous Dysfunction and Insulin Resistance in Pancreatic α Cells

Type 2 diabetes is caused by relative insulin hyposecretion and insulin resistance. In particular, decreased insulin secretion due to reduced pancreatic β cell mass is considered the leading cause of the onset and exacerbation of diabetes. However, in recent years, it has been surprisingly reported that the administration of streptozotocin (STZ), which destroys β cells and completely inhibits insulin secretion, does not increase blood glucose levels in glucagon deficient mice [10]. When STZ was administered to glucagon receptor-deficient mice, the blood glucose levels remain normal even when insulin secretion is suppressed [11]. Furthermore, it has been shown that blood glucose levels increase when a glucagon receptor is transiently introduced into the liver of glucagon receptor-deficient mice treated with STZ [12]. These reports indicate that glucagon plays a more important role than insulin in increasing blood glucose levels and support the “glucagonocentric hypothesis” proposed in 2012 [13]. That is, glucagon and GRs have extremely important roles in the glucose intolerance in diabetes, and the glucagon pathway as therapeutic targets has attracted attention.

In vivo, glucagon secretion from the pancreatic α cells is strictly controlled by insulin [14,15]. Furthermore, in type 2 diabetes patients, insulin resistance is observed in the liver, muscle, and adipose tissue, which are the classical insulin target organs. In recent years, insulin resistance in different organs in the body has also been proposed. Assuming that such insulin-resistant state is also present in pancreatic α cells, it is expected that insulin will not suppress glucagon secretion and that glucose tolerance will further aggravate (Figure 1). Therefore, identifying the regulatory mechanism of glucagon secretion and insulin resistance in pancreatic α cells can contribute to the elucidation of the pathophysiology of diabetes.

### 2.1. Glucose-Regulated Glucagon Secretion in α Cells (Cell Autonomous)

When the blood glucose level deviates from the normal level, either hypoglycemia or hyperglycemia occurs. When healthy individuals are subjected to fasting, insulin secretion from β cells is suppressed, whereas glucagon secretion from α cells is enhanced. By contrast, opposite changes are expected to occur when healthy individuals are fed. However, in patients with diabetes, a paradoxical glucagon hypersecretion occurs despite the presence of hyperglycemia. The most important factor affecting glucagon secretion in healthy individuals is glucose per se. Several glucose transporters, including GLUT1, sodium glucose co-transporter (SGLT1), and SGLT2, are expressed in human pancreatic α cells [16,17,18,19]. However, we recently reported that only SGLT1, not SGLT2, is expressed in both human and mouse pancreatic α cells [20]. In pancreatic α cells, glucose uptake occurs via the aforementioned GLUT, and the adenosine triphosphate (ATP)/adenosone diphosphate ratio increases via ATP production during glycolysis and tricarboxylic acid cycle (TCA cycle). In addition, ATP depolarizes K_ATP_ channels and reduces glucagon secretion [21]. ATP depolarizes K_ATP_ channels resulting in the influx of Na^+^ from Na^+^ channels, followed by Ca^2+^ influx via P/Q-type Ca^2+^ channels [22]. Although Ca^2+^ is the main trigger of glucagon secretion, under conditions where glucose strongly suppresses glucagon secretion in isolated islets, the Ca^2+^ signal is only moderately or temporarily decreased [23]. In addition, it has been reported that hypoglycemia stimulates glucagon secretion by directly elevating cyclic adenosine monophosphate (cAMP) concentrations regardless of Ca^2+^ signaling [24]. That is, these reports indicate that glucagon secretion is affected by various factors, and that Ca^2+^ and cAMP are only sufficient regulators for glucagon secretion. Under high glucose conditions, an SGLT-dependent increase in the intracellular Na^+^ concentration and a decrease in the mitochondrial pH increase glucagon secretion via the K_ATP_ channels [25]. These phenomena may partially explain the paradoxical glucagon hypersecretion in diabetes. As previously mentioned, several researchers have recently reported about the molecular mechanism of glucose-regulated glucagon secretion from pancreatic α cells, but such mechanism has not been fully elucidated yet. The mechanism is believed to involve K_ATP_ channels, such as insulin secretion and the cAMP-PKA signal. However, such a result must be validated in future studies.

### 2.2. Insulin Resistance in Pancreatic α Cells

In diabetes, paradoxical glucagon hypersecretion occurs despite the presence of hyperglycemia, and glucagon hyposecretion occurs despite the presence of hypoglycemia [26,27,28]. These phenomena lead to the worse of blood glucose and prolonged hypoglycemia. The pathologies indicate the dysfunction of pancreatic α cells and can be partially explained by insulin resistance in pancreatic α cells.

In general, insulin resistance is a condition in which classical insulin target organs have decreased insulin sensitivity resulting in decrease of glucose uptake [29]. Increased insulin resistance impairs the IR downstream signaling pathway, accompanied with decreased glucose uptake in the cells and increased blood glucose levels. Insulin is involved in blood glucose levels by glucose uptake in classical insulin target organ such as liver, adipose tissue and muscle. Insulin is also involved in blood sugar levels by suppressing glucagon secretion in pancreatic α cells. In a study of pancreatic perfusion, the administration of insulin suppressed glucagon secretion in dogs with alloxan-induced diabetes and in rats with STZ-induced diabetes [30]. Furthermore, the addition of insulin antibodies to human islets has resulted in the increase in glucagon secretion, and in sulfonylurea receptor (SUR)1 knockout mice that lack glucose-induced insulin secretion, glucagon secretion under high glucose conditions was not suppressed [31]. However, SUR1 is also expressed in pancreatic α cells and involved in glucagon secretion via activation of N-type Ca^2+^ channel [31]. This mechanism may also be involved in the enhancement of glucagon secretion in SUR1 knockout mice. Therefore, these reports have indicated that insulin suppress glucagon secretion. This effect may be mediated directly via the insulin receptor in α cells or indirectly via an increase in somatostatin secretion from pancreatic δ cells [32]. Insulin binds to the IR of α cells, inhibits the intracellular cAMP-PKA pathway, and reduces glucagon secretion in α cells [33]. In addition, some reports have shown that the sensitivity of K_ATP_ channels is reduced via PI3K, which is a signal molecule downstream of IR, to suppress glucagon secretion [34,35]. In addition, IR downstream Akt is involved in gamma-aminobutyric acid (GABA) receptor translocation to the cell membrane [36]. Therefore, the molecular mechanism underlying the suppression of glucagon secretion by insulin has been elucidated (Figure 2).

In diabetic state, insulin resistance might exist in pancreatic α cells. If insulin resistance is present in pancreatic α cells, these insulin signals are attenuated and affect glucagon secretion. In fact, studies of pancreatic α-cell-specific IR knockout mice have shown increased glucose and plasma glucagon levels ad libitum fed [15]. Furthermore, in the same mouse, impaired glucagon secretion during hypoglycemia was observed, and it partially mimics the phenomenon of diabetes [15]. In in vitro studies of glucagon secretion, rat-derived InR1G and mouse-derived αTC1 glucagonoma cell lines are commonly used. When InR1G cells are cultured for a long time under high glucose conditions, glucagon secretion and glucagon mRNA expression are enhanced based on glucose levels [37]. In addition, in the same cells, insulin reduces preproglucagon (PPG) mRNA expression [38]. However, in the presence of ceramide, which induces lipid toxicity, the insulin-induced decrease in PPG mRNA expression is inhibited [38]. Moreover, glucagon secretion is suppressed by insulin using αTC1 cells. By contrast, insulin did not suppress glucagon secretion in the presence of palmitic acid, which induces lipid toxicity [39]. Under such conditions, the phosphorylation of insulin receptor substrate (IRS), PI3K, and Akt, which are the downstream of IR, are suppressed, whereas the expression of paired box (PAX) 6, a key transcription factor of *proglucagon* gene, is enhanced via mitogen-activated protein kinase (P-38) and extracellular signal-regulated kinase (ERK) 44/42 pathways [39]. These reports indicate the glucagon paradoxical hypersecretion in diabetic state including lipotoxic condition. Furthermore, the knockdown of IR in αTC1 cells reduced glucagon secretion under low glucose conditions [40], which indicates that glucagon hyposecretion and hypoglycemia are prolonged in diabetic state. However, the increased glucagon secretion in hypoglycemia is mainly related to the sympathetic nerve. Pancreatic α-cell-specific IR knockout mice have shown the reduced norepinephrine response at basal and hypoglycemic conditions, suggesting interactions between hypoglycemia and sympathetic nerves under post insulin receptor signaling [41]. In addition, we need to consider the interactions among islet cells under hypoglycemic conditions. Therefore, the mechanism by which glucagon secretion is reduced in hypoglycemia is still controversial, although it may be partially explained by insulin resistance in pancreatic α cells.

## 3. Other Islet Cell Factors Regulating Glucagon Secretion

Paracrine signaling indicates that secretion from cells does not act on distant cells through the general circulation but acts on adjacent cells via direct diffusion and other similar mechanisms. In rodent islets, β cells are found in the center of the islet and α cells around the islets. Anatomically, paracrine factors from β cells may affect α cells, considering that blood flows from the central to the peripheral part in the islet [42]. By contrast, in human islets, islet cells are organized in a disorderly manner. However, human α cells also have β cells surrounding the blood vessels [43]. Therefore, it is possible to consider the influence of paracrine on pancreatic α cells from pancreatic β cells also in human.

Just as insulin from β cells suppresses glucagon secretion from α cells, glucagon secretion from pancreatic α cells is autocrinally and paracrinally regulated by various factors secreted by the pancreatic islet α, β, and δ cells. GABA, Zn^2+^, and insulin secreted by pancreatic β cells suppress glucagon secretion from α cells. GABA is known as a major inhibitory transmitter in the central nervous system, but has also been shown to be present with high concentrations in the pancreas [44,45,46]. In β cells, GABA is synthesized from glutamine via the action of glutamic acid decarboxylase (GAD) and is released when the decellularization of β cells occurs and the intracellular free Ca^2+^ concentration is increased. In mouse islets and α cell lines (α-TC1-9), GABA released from β cells binds to the GABA-A receptors of α cells and suppresses glucagon secretion [47,48,49]. Additionally, Zn^2+^ is contained in the insulin granules of pancreatic β cells. In the perfused pancreas of rats, glucose-induced Zn^2+^ secretion from β cells suppressed glucagon secretion [50]. However, in some reports that used mouse pancreatic islets, Zn^2+^ did not suppress glucagon secretion [51]. Furthermore, no changes were observed in glucose-induced glucagon secretion in Zn^2+^ granule transporter knockout mice [52]. Therefore, the contribution of Zn^2+^ in regulating glucagon secretion remains controversial. In addition, glucagon exocytosis in α cells is inhibited by juxtacrine via the Ephin subtype A (EphA) of β cells and EphA 4/7 receptor of α cells [53]. That is, glucagon secretion from pancreatic α cells is suppressed by paracrine and juxtacrine from pancreatic β cells. Somatostatin secreted by δ cells also suppresses glucagon secretion from α cells similar to that from β cells [54,55]. Somatostatin receptor (SSTR) subtype 2 is present in α cells which suppresses glucose-induced glucagon secretion by reducing intracellular cAMP levels [32,56,57]. Moreover, glucagon secretion increased in isolated islets of SSTR2 knockout mice [58]. Somatostatin inhibits glucagon secretion in the pancreatic α cell line InR1G9 cells [59]. Moreover, these reports have supported the notion that somatostatin suppresses glucagon secretion from α cells. The suppression of arginine-induced glucagon secretion was observed in systemic somatostatin knockout mice. However, it did not affect basal glucagon secretion [60]. In addition, in rats, the administration of SSTR2-specific antagonists that inhibited insulin secretion with STZ treated mice did not alter blood glucagon levels [61]. That is, somatostatin suppresses glucagon secretion. However, somatostatin alone cannot completely suppress glucagon secretion. In addition, GRs are present in pancreatic α cells [62,63,64]. Western blot and immunohistological staining confirmed the presence of GRs in human, mouse pancreatic islets and α cell lines (α-TC1-9) [65]. Glucagon secreted from α cells binds to its own GR, promotes its own glucagon secretion via the cAMP-PKA pathways, and up-regulates its own gene expression in human and mouse islets and α cell line (α-TC1-9). In addition, the expression of Gcg mRNA decreased when glucagon receptor antagonists were added to mouse islets and α-TC1-9 cells [65]. Therefore, glucagon secretion in pancreatic α cells is controlled by other islet cells or themselves, and such mechanism is associated with the pathogenesis of diabetes (Figure 3).

## 4. Diabetes Therapy Targeting Glucagon

Currently, there are several different diabetes drugs available worldwide, some of which affect glucagon in various ways and exhibit hypoglycemic action. Glucagon-like peptide-1 (GLP-1) agonist and dipeptidyl peptidase-4 (DPP4) inhibitor, which are incretin-related drugs, suppress glucagon secretion. DPP4 is an enzyme that inhibits GLP-1. When either of these drugs is administered, the concentration of GLP-1 in the blood increases. Administration of GLP-1 suppresses glucagon secretion in patients with type 2 diabetes, as measured using a glucose clamp test [66]. The mechanism by which GLP-1 suppresses glucagon reportedly involves the activation of the cAMP-PKA pathway via the GLP-1 receptors of pancreatic α cells and suppression of voltage-gated N-type Ca channel activity, thereby resulting in the suppression of glucagon secretion [67]. However, this mechanism is controversial, because the GLP-1 receptor is only expressed at low levels in pancreatic α cells. On the other hand, an indirect mechanism in which GLP-1 induces somatostatin secretion from pancreatic δ cells and suppresses glucagon secretion via the SSTR-2 of pancreatic α cells has been reported [68]. GLP-1 increases insulin secretion from β cells and suppresses glucagon secretion. Therefore, although several mechanisms for the suppression of glucagon secretion by GLP-1 have been reported, the details have not yet been completely elucidated [69]. On the other hand, biguanide has an inhibitory effect on glucagon. Metformin, a member of the biguanide class of antidiabetic drugs, improves insulin resistance by suppressing gluconeogenesis in the liver and promoting glucose uptake in the skeletal muscle. In the liver, metformin suppresses complex I in the mitochondria. The main mechanism of action is considered to be the suppression of gluconeogenesis and release of glucose via the activation of AMP-activated protein kinase resulting from the elevation of AMP concentrations [70]. However, recently it has been reported that glucagon inhibition by metformin results from the elevation of AMP concentration in the liver, which promotes the inhibition of adenylate cyclase via glucagon receptors and suppresses the cAMP-PKA pathway, thereby leading to the suppression of glucose production by glucagon [71]. By contrast, other antidiabetic drugs-sulfonylurea, glinide, and SGLT2 inhibitor are considered to promote glucagon secretion to different extents.

In addition, glucagon receptor antagonists that suppress gluconeogenesis in the liver and lower blood glucose levels have recently been developed. LY2409021 is a low molecular weight compound that is an antagonist to the glucagon receptor. A marked improvement in blood glucose was observed after 12 weeks of administration of this compound to patients with type 2 diabetes [72]. However, the development of LY2409021 has recently been stopped due to side effects such as weight gain, blood pressure increase, and fatty liver disease [73,74]. Moreover, neutralizing antibodies against glucagon receptors have recently been developed. REND 2.59, a neutralizing antibody against glucagon receptors, has shown remarkable improvement in blood glucose levels in studies using diabetic model mice [75]. There were no side effects observed with the use of LY2409021, and weight loss as well as improved fatty liver disease were observed [75].

Among various antidiabetic drugs currently available, incretin-related drugs and biguanide are considered to exert a glucagon inhibitory effect, and their mechanism of action has been elucidated. Furthermore, new drugs including glucagon receptor antagonists and neutralizing antibodies are currently under development, and the importance of glucagon in diabetes is increasingly recognized.

## 5. Limitations of Pancreatic α Cell Research

Studies regarding to the glucagon are few compared to those of insulin, and there are two major reasons. First, there is a problem with the glucagon assay system. Conventionally, glucagon is measured by using radioimmunoassay methods with antibodies that recognize the C-terminal structure of glucagon. However, these methods cannot accurately measure glucagon because of the existence of glucagon-like peptides, whose amino acids sequence overlap with glucagon. In particular, glicentin, the C-terminal sequence of which is identical to that of glucagon, causes cross-reaction. However, in recent years, glucagon can be more accurately measured by newly developed methods, i.e., sandwich enzyme-linked immunosorbent assay using antibodies that recognize the C-terminus and N-terminus of glucagon, and liquid chromatography/mass spectrometry (LC/MS/MS). Therefore, the problem seems to be solved (but this is not the case). Second, there is a problem with the material used. In the case of mice and rats used in the experiment, islets cells are mainly composed of β cells and α cells by around 80% and 10%, respectively. Therefore, α cell research is challenging because the number of α cells is much fewer than β cells. In addition, although insulin is a relatively stable hormone, glucagon is unstable and rapidly broken down at room temperature. Moreover, most studies analyzing glucagon secretion have used transformed α cell lines, and only a few studies have utilized native α cells, which is attributed to the absence of established methods that isolate pancreatic α cells owing to their unstable viability. Currently, the commonly used α cell lines are InR1G9 [76], HIT T15-G [77], and α-TC1 [78] cells. However, the ability of the first two cell lines to mimic native pancreatic α cells is debatable because they were initially derived from insulinomas (β cells). Although α-TC1 cells are derived from glucagonomas, glucagon secretion in response to glucose and insulin has been poor because oncogene is constantly expressed; thus, the cells, such as native α cells, are in the growth phase and do not exhibit differentiation characteristics. Therefore, the establishment of new methods for isolating native α cells or a new α cell line with similar properties to native α cells is required in α cell research.

## 6. Conclusions

In type 2 diabetes, glucagon dysregulation is as important as insulin dysregulation. In fact, paradoxical glucagon hypersecretion under hyperglycemic conditions and impaired glucagon secretion under hypoglycemic conditions are commonly observed in diabetes. Therefore, the relationship between insulin and glucagon—the major factors that determine blood glucose level—is important in diabetes research. In recent years, insulin resistance in pancreatic α cells has been proposed; moreover, the relationship between insulin and glucagon has been gaining considerable attention. In insulin-resistant α-cell lines, the regulation of glucagon secretion and expression is disturbed. In pancreatic α cell-specific IR knockout mice, increased postprandial plasma glucagon levels and hyperglycemia have been observed. Therefore, insulin resistance in α cells is associated with glucagon dysregulation in diabetes. Although the mechanism underlying cell autonomous dysfunction of α cells has not yet been elucidated, it is believed to be correlated with the pathophysiology of diabetes. Therefore, α cell research using native α cells or α cell lines with similar properties to native α cells should be performed for better understanding of the pathophysiology of diabetes.

Here are the final summaries;
Type 2 diabetes is considered to be a “bi-hormonal disorder” rather than an “insulin-centric disorder,” suggesting that glucagon is as important as insulin. Recently, “glucagonocentric hypothesis”, in which glucagon contributes more to increase in blood sugar level than insulin, has attracted attention.Insulin resistance in pancreatic α cells means a state in which the insulin signal of α cells is attenuated. In normal conditions, insulin suppresses the secretion of glucagon from pancreatic α cells. However, when insulin resistance exists in diabetic pancreatic α cells, insulin can no longer suppress glucagon secretion from α cells, which results in hypersecretion of glucagon. Hypothetical paradoxical glucagon hypersecretion observed in diabetes.The molecular mechanism of glucose-regulated glucagon secretion in pancreatic α-cells has not yet been elucidated, but the involvement of cAMP, ATP/ADP ratio, and K_ATP_ channel has been reported.The new drugs such as glucagon receptor antagonist and neutralizing antibody are currently developing.Glucagon studies have problems with measurement systems and materials. The problem of the measurement system is being solved, but the problem of the material has not been solved yet. Therefore, the establishment of new methods for isolating native α cells or construction of a new α cell line having similar properties with native α cells is required for future α cell research.

## Figures and Tables

**Figure 1 ijms-20-03699-f001:**
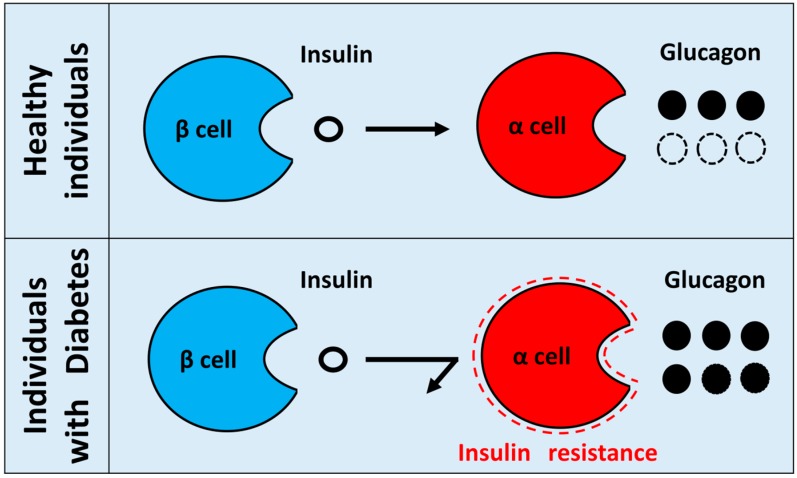
Hypothesis of insulin resistance regulating glucagon secretion in pancreatic α cells. In healthy individuals, insulin from pancreatic β cells regulates glucagon secretion from pancreatic α cells. By contrast, when insulin resistance exists in diabetic pancreatic α cells, insulin can no longer suppress glucagon secretion from α cells, which results in hypersecretion of glucagon. Hypothetical paradoxical glucagon hypersecretion observed in diabetes.

**Figure 2 ijms-20-03699-f002:**
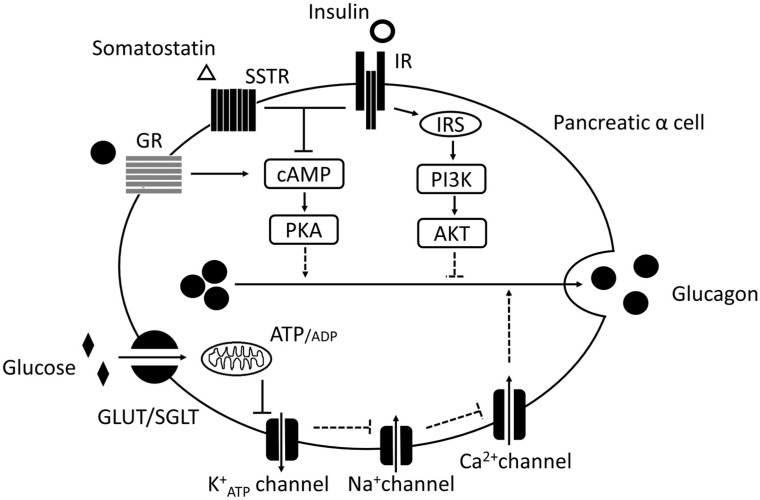
Hypothesis of the molecular mechanisms in pancreatic α cells. In pancreatic α cells, glucose translocates via glucose transporters, such as GLUT or SGLT. Glucose suppresses glucagon secretion via adenosine triphosphate (ATP) production in the glycolytic and tricarboxylic acid cycle. K^+^ATP, Na^+^, and Ca^2+^ channels are involved in this molecular mechanism. Moreover, insulin suppresses glucagon secretion via IR, IRS, PI3K, and Akt. Somatostatin suppresses glucagon secretion via somatostatin receptor (SSTR) and cAMP-PKA pathways. Glucagon enhances glucagon secretion via glucagon receptor (GR) per se and cAMP-PKA pathways. Solid arrow: canonical positive signal, dotted arrow: hypothetical positive signal, solid T arrow: canonical negative signal, dotted T arrow: hypothetical negative signal.

**Figure 3 ijms-20-03699-f003:**
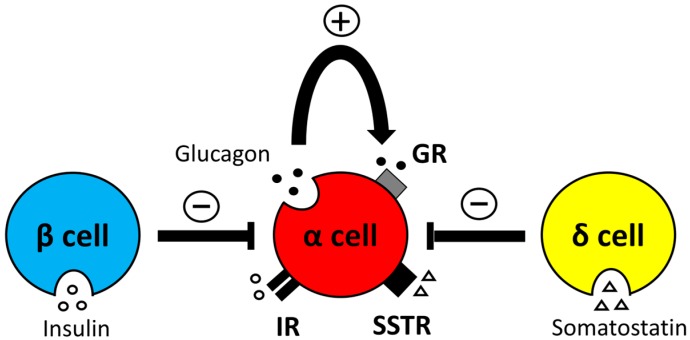
Relationship between islet hormones and glucagon secretion in pancreatic α cells. Insulin, glucagon, and somatostatin subtype 2 receptors are present in α cells. Glucagon secretion in α cells is suppressed by somatostatin and insulin secreted by β and δ cells, respectively. Meanwhile, glucagon secreted by α cells per se enhances glucagon secretion. “+”: Enhancement of glucagon secretion. “−”: Suppression of glucagon secretion.

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
