# Peer review of "Cell Autonomous Dysfunction and Insulin Resistance in Pancreatic α Cells"

_ijms, 2019, doi:10.3390/ijms20153699_

Round 1
Reviewer 1 Report
This is a nice review by Honzawa et al. where the importance of insulin resistance in α-cell on glycemic control is discuss. Noteworthy, the authors include a section where they do an interesting discussion about the constrains in α cell research. The authors have done an excellent job organizing a complex topic in a manner that is informative and approachable, however some issues need to be address:
Major:
-This review points out the role of glucagon in T2D, regarding this it would be interesting to include a section where the authors discuss available drugs/possible treatments based on glucagon as therapeutic target.
-The review would be benefit of proofreading by an English speaker.
Minor:
- Page 3 line 91-93: Please rephrase this sentence as it’s a bit confusing: “In addition, Ca2+ signaling is a permissive action because it does not necessarily change Ca2+ signal under glucagon secretion conditions where the level of cAMP is elevated”
Page 5 line 167: Please rephrase this sentence as it’s a bit confusing: “It is also known to be present in high concentrations in the pancreas, and GABA is a major inhibitory transmitter in the central nervous system”. Indicate that it refers to GABA at the beginning of the sentence.
-Figure legend 2. Please briefly explain SSTR and GR role in glucagon release.
-Parts of the review lack of references.
Author Response
RESPONSE TO REVIEWER #1
Major
Comment 1:
This review points out the role of glucagon in T2D, regarding this it would be interesting to include a section where the authors discuss available drugs/possible treatments based on glucagon as therapeutic target.
Response: Thank you for your comment. Accordingly, we have added pertaining to available drugs/possible treatments based on glucagon as therapeutic target as following.
"Diabetes therapy targeting glucagon" to the text (page 5-6, line 202-242).
Minor
Comment 2:
Page 3 line 91-93: Please rephrase this sentence as it’s a bit confusing: “In addition, Ca2+ signaling is a permissive action because it does not necessarily change Ca2+ signal under glucagon secretion conditions where the level of cAMP is elevated”
Response: We totally agreed with your comment. Thus, we have changed as following.
“In addition, it has been reported that hypoglycemia stimulates glucagon secretion by directly elevating cAMP concentrations regardless of Ca2+ signaling [24]. That is, these reports indicate that glucagon secretion is affected by various factors, and that Ca2+ and cAMP are only sufficient regulators for glucagon secretion. ” (page 3, line 91-94).
Comment 3:
Page 5 line 167: Please rephrase this sentence as it’s a bit confusing: “It is also known to be present in high concentrations in the pancreas, and GABA is a major inhibitory transmitter in the central nervous system”. Indicate that it refers to GABA at the beginning of the sentence.
Response: Thank you for your pivotal comment. We have simplified it as follows. “GABA is known as a major inhibitory transmitter in the central nervous system but is also present in high concentrations in the pancreas.” (page 5, line 169-171).
Comment 4:
Figure legend 2. Please briefly explain SSTR and GR role in glucagon release.
Response: Accordingly, we have added a brief explanation of SSTR and GR as follows. “Figure 2. Hypothesis of the molecular mechanisms in pancreatic a cells. In pancreatic a cells, glucose translocates via glucose transporters, such as GLUT or SGLT. Glucose suppresses glucagon secretion via adenosine triphosphate (ATP) production in the glycolytic and tricarboxylic acid cycle. K+ATP, Na+, and Ca2+ channels are involved in this molecular mechanism. Moreover, insulin suppresses glucagon secretion via IR, IRS, PI3K, and Akt. Somatostatin suppresses glucagon secretion via SSTR and cAMP-PKA pathways. Glucagon enhances glucagon secretion via GR per se and cAMP-PKA pathways.” (page 8, Figure 2, line 280-285).
Comment 5:
Parts of the review lack of references
Response: Thank you for your advice. Some references have been added and deleted also referring to comments of other reviewers. I would appreciate your confirmation (page 11-15, line 347-557).
Deleted
Taborsky, G. J., Jr.; Ahren, B.; Mundinger, T. O.; Mei, Q.; Havel, P. J. Autonomic mechanism and defects in the glucagon response to insulin- induced hypoglycaemia. Diabetes Nutr. Metab. 2002, 15, 318–322.
Added
Ref 27, 28, 35, 68-77.
Thank you again for your constructive comments.
We hope revised manuscript is now suitable for publication.
Reviewer 2 Report
In this review article, the authors describe the dysregulation of alpha cell function and how it leads to inappropriate glucagon secretion, thereby contributing to aberrant glycaemia in diabetes. They primarily cover nutrient regulation and inter-islet cell communications with alpha cells, and provide minimal coverage on the involvement of sympathetic regulation. The review is well written, although not novel. But it is still arguable at several places.
1. In line 48, the statement of ghrelin secretion from alpha cells is not likely accurate.
As previously shown, islet ghrelin producing cells only appear at the developmental stage in mice, rat and human. However, ghrelin cells quickly decrease after birth, and it is very difficult to found them in adult mice. In addition, although, in mice, a large proportion of ghrelin cells in developing pancreas harbors glucagon, it has been reported that human islet ghrelin cells have no glucagon expression at any stage of development.
2. In line 58, ‘alpha cell deficient mice’ should be replaced with ‘glucagon deficient mice’
3. In line 66, ref 14 does not fit to this context, because it discussed the possibility that an autonomic defect contributes to the loss of the glucagon response to insulin-induced hypoglycemia in T1DM.
4. In line 93-94, the text needs to be able to be understood as a context of ref 25, in which it is described that hypoglycemia stimulates glucagon secretion by elevating the cAMP concentration.
5. In line 105, is attenuated glucagon response to hypoglycemia in diabetes implicated with reduced insulin actions in alpha cells? In addition, ref 27 is very old. The authors may want to put more recent papers.
6. In line 117, knockout mice which lacking ~ →knockout mice that lack
7. The mechanisms underlying dysregulated glucagon secretion in SUR1 KO mice are complicated. Not only paucity of insulin but also lack of SUR1 itself in alpha cells could be involved in dysregulation of glucagon secretion. The author could provide expanded explanation on that.
8. Does ref 33 correspond to the text in line 122-123?
9. In line 126-130, and in line 137
I think that this concept is yet to be conclusive. Conversely, the recent papers including ref 16 do not support it. The beta and alpha cell interaction is bidirectional and is more complicated. As evidenced in recent papers, glucagon itself has direct actions in beta cells and stimulates insulin secretion. On the other hands, islet hormone secretion in the presence of hypoglycemia is more likely dependent on sympathetic nerve input. In addition, in ref 16, it is discussed that impaired glucagon response to hypoglycemia in the setting of IR deficiency in alpha cells is associated with altered sympathetic tone. This suggests that IR signaling in alpha cells has roles on the regulation of sympathetic tone. The authors need to more carefully discuss physiological and pathophysiological roles of glucagon.
10. All picture are nicely drawn.
Author Response
RESPONSE TO REVIEWER #2
Comment 1:
In line 48, the statement of ghrelin secretion from alpha cells is not likely accurate.
As previously shown, islet ghrelin producing cells only appear at the developmental stage in mice, rat and human. However, ghrelin cells quickly decrease after birth, and it is very difficult to found them in adult mice. In addition, although, in mice, a large proportion of ghrelin cells in developing pancreas harbors glucagon, it has been reported that human islet ghrelin cells have no glucagon expression at any stage of development.
Response: We totally agree with your comment. Thus, we have corrected as following.
”ghrelin (secreted from pancreatic α cells)” in this article, correctly it is” ghrelin (secreted from pancreatic ε cells)” (page 2, line 48). Also, thank you for teaching us about the differences between ghrelin cells and glucagon expression in mice and humans at the developmental stage.
Comment 2:
In line 58, ‘alpha cell deficient mice’ should be replaced with ‘glucagon deficient mice’.
Response: Thank you for your comment. We have corrected it to ‘glucagon deficient mice’ (page 2, line 58).
Comment 3:
In line 66, ref 14 does not fit to this context, because it discussed the possibility that an autonomic defect contributes to the loss of the glucagon response to insulin-induced hypoglycemia in T1DM.
Response: We totally agree with your comment. Thus, we have removed the Ref 14 in this article.
Comment 4:
In line 93-94, the text needs to be able to be understood as a context of ref 25, in which it is described that hypoglycemia stimulates glucagon secretion by elevating the cAMP concentration.
Response: Thank you for your comment. According, we have corrected it to “In addition, it has been reported that hypoglycemia stimulates glucagon secretion by directly elevating cAMP concentrations regardless of Ca2+ signaling [24]. That is, these reports indicate that glucagon secretion is affected by various factors, and that Ca2+ and cAMP are only sufficient regulators for glucagon secretion.” (page 3, line 91-94).
Comment 5:
In line 105, is attenuated glucagon response to hypoglycemia in diabetes implicated with reduced insulin actions in alpha cells? In addition, ref 27 is very old. The authors may want to put more recent papers.
Response: Thank for your question. It may be partially explained by α cell insulin resistance. Pancreatic α-cell-specific IR knockout mice show low response to norepinephrine in basal and hypoglycemic states, suggesting a relationship between hypoglycemia and insulin and sympathetic nerves. If insulin resistance is present in α cells, this sympathetic nerve response is reduced and pancreatic α cell glucagon secretion is suppressed. However, the influence of switch-off mechanism and adjacent cell interaction is also considered, which is controversial.
In addition, thank you for your assessment of ref 27. The following paper has been added. I hope you can confirm (page 12, line 416-419).
Unger, R. H.; Orci, L. The essential role of glucagon in the pathogenesis of diabetes mellitus. Lancet 1975, 1, 14-16. (ref 27)
Cryer, P. E. Mechanisms of hypoglycemia-associated autonomic failure in diabetes. N Engl J Med 2013, 369, 362-372. (ref 28)
Comment 6:
In line 117, knockout mice which lacking ~ →knockout mice that lack
Response: We have corrected it to “knockout mice that lack” (page 3, line 118).
Comment 7:
The mechanisms underlying dysregulated glucagon secretion in SUR1 KO mice are complicated. Not only paucity of insulin but also lack of SUR1 itself in alpha cells could be involved in dysregulation of glucagon secretion. The author could provide expanded explanation on that.
Response: Thank you for your assessment. Certainly, we should explain the action of SUR1 on α cells. Therefore, we have changed to the following.
“Furthermore, the addition of insulin antibodies to human islets has resulted in the increase in glucagon secretion, and in sulfonylurea receptor (SUR)1 knockout mice that lack glucose-induced insulin secretion, glucagon secretion under high glucose conditions was not suppressed [31]. However, SUR1 is also expressed in pancreatic α cells and is reported to be involved in glucagon secretion via activation of N-type Ca 2+ channel [31]. This mechanism may also be involved in the enhancement of glucagon secretion in SUR1 knockout mice.” I hope you can confirm (page 3, line 119-121).
Comment 8:
Does ref 33 correspond to the text in line 122-123?
Response: Thank you for your comment. This reference alone did not indicate the content of the sentence. Therefore, the following paper has been added (page 4, line 127).
Leung, Y. M.; Ahmed, I.; Sheu, L.; Gao, X.; Hara, M.; Tsushima, R. G.; Diamant, N. E.; Gaisano, H. Y.; Insulin regulates islet alpha-cell function by reducing KATP channel sensitivity to adenosine 5'-triphosphate inhibition. Endocrinology 2006, 147, 2155-2162. (ref 35)
Comment 9:
In line 126-130, and in line 137
I think that this concept is yet to be conclusive. Conversely, the recent papers including ref 16 do not support it. The beta and alpha cell interaction is bidirectional and is more complicated. As evidenced in recent papers, glucagon itself has direct actions in beta cells and stimulates insulin secretion. On the other hands, islet hormone secretion in the presence of hypoglycemia is more likely dependent on sympathetic nerve input. In addition, in ref 16, it is discussed that impaired glucagon response to hypoglycemia in the setting of IR deficiency in alpha cells is associated with altered sympathetic tone. This suggests that IR signaling in alpha cells has roles on the regulation of sympathetic tone. The authors need to more carefully discuss physiological and pathophysiological roles of glucagon.
Response: Thank you for your assessment. We agree with your opinions. Certainly, the mechanisms of the increased glucagon secretion in hypoglycemia are considered to be complex. Although the mechanisms are mainly related to sympathetic nerves, as you pointed out, we should consider the interaction among islet cells and the switch-off mechanism. Therefore, I deleted the line 126-130 (pre-revision, the explanation of switch-off mechanism) and corrected the paragraph “Insulin resistance in pancreatic α cells” a little (page 3-4, line 103-155). I hope you can check the red part of the revision article. Kind regard.
Comment 10:
All picture are nicely drawn.
Response: We are glad that you like the drawings.
Many thank you for your constructive comments.
We hope that the revised manuscript is now suitable for publication.
Round 2
Reviewer 1 Report
All questions have been addressedReviewer 2 Report
The manuscript was sufficiently revised. I agree that it is going to be published.